# Accuracy is Not Enough: Evaluating Personalization in Summarizers

**Rahul Vansh** [†*]  **Darsh Rank** [†*]  **Sourish Dasgupta** [†*]  **Tanmoy Chakraborty** [‡*]

† KDM Lab, Dhirubhai Ambani Institute of Information & Communication Technology, India

‡ Indian Institute of Technology, Delhi, India

{202111035, 201901247, sourish_dasgupta}@daiict.ac.in, tanchak@iitd.ac.in

## Abstract

Text summarization models are evaluated in terms of accuracy and quality using various measures such as ROUGE, BLEU, METEOR, BERTScore, PYRAMID, readability, and several recently proposed ones. The central objective of all accuracy measures is to evaluate the model's ability to capture *saliency* accurately. Since saliency is subjective w.r.t the readers' preferences, there cannot be a fit-all summary for a given document. This means that in many use cases, summarization models need to be personalized w.r.t user profiles. However, to our knowledge, there is no measure to evaluate *the degree of personalization* of a summarization model. In this paper, we first establish that existing accuracy measures cannot evaluate the degree of personalization of any summarization model and then propose a novel measure, called EGISES, for automatically computing the same. Using the PENS dataset released by Microsoft Research, we analyze the degree of personalization of ten different state-of-the-art summarization models (both extractive and abstractive), five of which are explicitly trained for personalized summarization, and the remaining are appropriated to exhibit personalization. We conclude by proposing a generalized accuracy measure, called *P-Accuracy*, for designing accuracy measures that should also take personalization into account and demonstrate the robustness and reliability of the measure through meta-evaluation.

## 1 Introduction

The growing availability of large-scale text data has led to an increasing demand for automated text summarization systems that aim to compress a lengthy document into a short paragraph, which includes salient information about the document. Evaluation of such summarization systems (and the underlying models) is performed either in terms of their accuracy or quality. To date, several accuracy

---

*All authors have equal contributions.

measures have been proposed such as the widely adopted ROUGE variants (e.g., ROUGE-*n*/L/SU4 etc.) (Lin, 2004), BLEU (Papineni et al., 2002), METEOR (Banerjee and Lavie, 2005), BERTScore (Zhang et al., 2019), PYRAMID (Nenkova and Passonneau, 2004; Gao et al., 2019) and the more recently proposed ones such as SUPERT (Gao et al., 2020), WIDAR (Jain et al., 2022), and InfoLM (Colombo et al., 2022). At the same time, several meta-evaluation studies on the reliability of these measures based on the human judgment correlation have been proposed (Graham, 2015; Chatzikoumi, 2020; Bhandari et al., 2020; Deutsch et al., 2021; Fabbri et al., 2021; Peyrard, 2019; Wei and Jia, 2021).

Within the context of all these developments on novel metrics and meta-evaluation showing varied results on different datasets and summarization systems, the central idea remains the same for all accuracy measures, i.e., they all need to measure the ability of summarization models to capture *saliency* accurately. However, saliency is subjective to a reader's prior reading history and evolving preferences (*aka* attention drift). In other words, in many use cases, models have to consider user reading patterns and then generate personalized summaries instead of a fit-all generic summary. This calls for summarizers to be personalized. Although, relatively speaking, personalized summarization needs much research attention, there have been a few noteworthy studies in this direction (Ghodratnama et al., 2020b; Ao et al., 2021). The evaluation of these models has been mostly based on the accuracy measures (e.g., ROUGE-L). However, in this paper, we establish that we cannot measure the *degree of personalization* of summarization models using accuracy. We theoretically prove that personalization is an independent characteristic compared to accuracy, and it is possible that while a model performs fairly reasonably w.r.t any accuracy measure, it may have a poor degree of personalization.

We, thereby, propose a novel measure, called the *effective Degree of Insensitivity w.r.t Subjectivity* (EGISES) that conversely measures how insensitive a model is if there is a significant difference in the preferences (or expected summaries) of two readers for the same original document, where an insensitive model would not have much change in the generated summaries for the two readers. To the best of our knowledge, this notion of degree of personalization and its corresponding measure would be the first of its kind. We evaluate EGISES on ten state-of-the-art summarization models (both abstractive and extractive), including the best-performing ones proposed by (Ao et al., 2021) using the PENS test dataset (Ao et al., 2021), and show that the leaderboard so generated does not have any consistent correlation with those generated by standard accuracy measures, thereby empirically showing that accuracy is not enough.

A key step for any new accuracy metric is to establish its robustness and reliability. The standard method to do so is human-judgment-based meta-evaluation using correlation statistics such as Pearson's Coefficient, Spearman Rank Correlation, and Kendall $\tau$ Rank Correlation. Since direct methods are practically infeasible (we will discuss that in Section 6.1) we adopt an indirect meta-evaluation method that establishes a high human-judgment vs. EGISES correlation. We then propose a generic accuracy measure based on EGISES, called $P\text{-}Accuracy$, for calculating the realistic accuracy of models that need to exhibit personalization, and have empirically shown that the measure is stable w.r.t its invariance to the original accuracy leaderboard.

## 2   Preliminaries

### 2.1   Degree of Personalization

The degree of personalization of a summarization model is the quantitative measure of the extent to which a model can generate summaries that align with the reader's subjective appreciation of saliency. While we provide a formal definition in Section 2.3, informally, if two readers' subjective agreement of saliency of a given document is low, then the model should generate personalized summaries for the readers such that they should not have high overlap, and vice-versa, without affecting the required accuracy. Although the general intuition might be that accuracy measures should be sufficient to measure this subjectivity, we demonstrate

that measuring personalization is completely different from measuring how close a model's generated summary is to that of the expected summary (which accuracy measures do). To elucidate this counter-intuitive aspect with an example, imagine Alice and Bob are interested in news related to the Russia-Ukraine war, but Bob is more focused on news related to the war fronts, while Alice is interested in news related to civilian distress. Now, for a given news article, a model may be able to generate a fairly accurate summary for Bob where the core narrative is about the battles, while at the same time, it is also able to get a good accuracy for Alice because it happens to have sufficient peripheral content (i.e., off-topic mentions) about, say, civilian displacement. In this situation, Alice has to filter out content that is not salient to her so as to get to her interest. This becomes even more problematic in the case when models also generate headlines. In our example, Alice will not get to see the summary headline aligned with her interest and, therefore, may completely skip the summary. Therefore, the degree of personalization provides an insight into how engaging a model can be in terms of better user experience and not just a clinical computation of the overlap of model-generated summaries to that of their gold references. In the following section, we will provide a mathematical sketch-of-proof that will clearly demonstrate that accuracy measures are not capable of measuring the degree of personalization of a model.

### 2.2   Accuracy is Not Enough - Proof Sketch

In this section, we provide a sketch-of-proof that the accuracy measures are insufficient to capture the degree of personalization of summarization models. For a given document $d_i$, two reader-profiles (i.e., expected personalized summaries or reference summaries) $u_{ij}$ and $u_{ik}$, and the corresponding summaries $s_{u_{ij}}$ and $s_{u_{ik}}$ generated by a model $M_{\boldsymbol{\theta},u}$, let's choose an arbitrary distance metric $\sigma$ defined on the metric space $M$ where $d, u, s$ are defined[1]. In other words, $\sigma(s, u)$ is the abstraction over any arbitrary accuracy measure. In a similar way, the notion of degree-of-personalization can be captured by the ratio $\frac{\sigma(u_{ij}, u_{ik})}{\sigma(s_{u_{ij}}, s_{u_{ik}})}$. We term this ratio as the *deviation*.

**Theorem 1.** *The deviation of a model $M_{\boldsymbol{\theta},u}$ on the metric space $M$ can be changed without any*

---

[1]$\sigma$ satisfies positivity, reflexive, maximality, symmetry, and the triangle inequality.

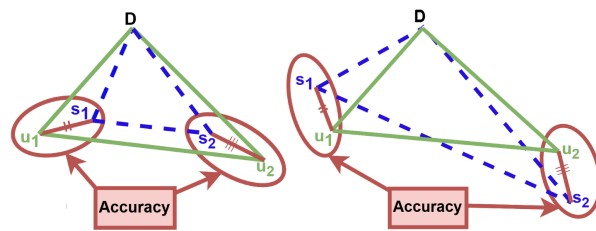

Figure 1: Triangulation: (a) High accuracy vs. low personalization; (b) High (and same) accuracy vs. high personalization; blue denotes *summary-deviation* and green denotes *reader-deviation*.

change in $\sigma(s, u)$.

*Proof.* Let $d, u, s$ be triangulated as per Figure 1. Keeping $d, u$ fixed, we can perform an arbitrary rotation operation ($rot(\bullet, u, \alpha)$; $\alpha$: angle of rotation) on $s_{u_{ij}}$ and $s_{u_{ik}}$ s.t. $rot(\bullet, u, \alpha)$ is a closure operator in $M$. Now, $\exists (p, q) \in M$, s.t.

$$\max_{(p,q)}\{\sigma(rot(s_{u_{ij}}, u_{ij}, \alpha_p), rot(s_{u_{ik}}, u_{ik}, \alpha_q))\} >$$

$$\min_{(p,q)}\{\sigma(rot(s_{u_{ij}}, u_{ij}, \alpha_p), rot(s_{u_{ik}}, u_{ik}, \alpha_q))\}$$

In other words, a total ordering of deviations exists. Also, for any arbitrary $\alpha$, $\sigma(u, rot(s, u, \alpha)) = \sigma(s, u)$ by the property of the rotation operator. Since we have kept $u$ fixed, the deviation, therefore, can be varied by changing $\alpha$ (and thereby $\sigma(s_{u_{ij}}, s_{u_{ik}})$) from a minimum to a maximum. $\square$

The proof shows that two models may have the same accuracy but different degree-of-personalization. Our findings support the same (Section 5.3). Therefore, accuracy measures are not adequate for measuring the degree-of-personalization. We term this proof framework as the *triangulation framework* and will continue to use this in our further discussions.

## 2.3 Measuring Insensitivity to Subjectivity

One way of looking at the degree of personalization is how insensitive a summarization model is in detecting the divergence in the subjectivity of readers' interests and preferences. If the model is highly insensitive, it means it practically generates almost the same summary irrespective of the divergence in the readers' subjective expectations. Hence, in such a case, the degree of personalization will be low and may lead to low engagement. In this section, we provide a formal and generic framework for designing measures for the degree of personalization in terms of *insensitivity-to-subjectivity*.

**Definition 1.** *Personalized Summarization Model.* *Given document $d$, and user profile $u$, a summarization model $M_{\boldsymbol{\theta}, u}$ is said to be personalized iff $M_{\boldsymbol{\theta}, u} \mapsto \widehat{s_u}$, where $\widehat{s_u}$ is the best estimated summary of $d$ considering user profile $u$, $\theta$ being model parameters.*

**Definition 2.** *Weak Insensitivity-to-Subjectivity.* *Given two mutually indistiguishable reader profiles, $u_i$ and $u_j$, a summarization model $M_{\boldsymbol{\theta}, u}$ is (weakly) Insensitive-to-Subjectivity iff $\forall (u_i, u_j), \ni \quad f^U_{dist}(u_i, u_j^*) \leq \tau^U_{max}$: $f^S_{sim}(M_{\boldsymbol{\theta}, u}(d, u_i), M_{\boldsymbol{\theta}, u}(d, u_j)) > \tau^S_{min}$, where $f^U_{dist}$ indicates user profile distance function, $f^S_{sim}$ is a summary similarity function, $\tau^U_{max}$ is the maximum limit for two different user profiles to be mutually indistinguishable, and $\tau^S_{min}$ is the minimum limit for two generated summary w.r.t two different users to be mutually distinguishable. $*$: $f^U_{dist}(u_i, u_i) = 0$ ; $f^U_{dist}(u_i, u_j) \in [0, 1]$.*

**Definition 3.** *Strong Insensitivity-to-Subjectivity.* *Given two different reader profiles, $u_i$ and $u_j$, a summarization model $M_{\boldsymbol{\theta}, u}$ is (strongly) Insensitive-to-Subjectivity iff the model satisfies the condition of weak insensitivity and also $\forall (u_i, u_j), \ni \quad f^U_{dist}(u_i, uj) > \tau^U_{max}$: $f^S_{sim}(M_{\boldsymbol{\theta}, u}(d, u_i), M_{\boldsymbol{\theta}, u}(d, u_j)) < \tau^S_{min}$.*

## 3 EGISES: Measure for Insensitivity

### 3.1 Triangulation Space Representation

To calculate how insensitive (or inversely, sensitive) a model is to the change in the readers' profiles when the same document has to be summarized for two different readers, we need to have a well-defined notion of *deviation* w.r.t an algebraic space where document set **D**, the model-generated summary set **S**, and readers' subjective expectation set **U** can be represented. In this paper, we make the *bag-of-word* assumption of $d \in \mathbf{D}$, $s \in \mathbf{S}$, and $u \in \mathbf{U}$, and represent them on the probability space for a specific document $d_j$ on the lines of the triangulation framework (see Section 2.2) as $(\Omega_{d_j}, \mathcal{F}_{d_j}, \mathbb{P})$:

$$\Omega_{d_j} = \{w_i \in \{d_j \bigcup_{k=1}^{|\mathbf{U}|} (u_{jk} \cup s_{u_{jk}})\} | w_i \in \mathbf{V}\} \quad (1)$$

$$\mathcal{F}_{d_j} = 2^{\Omega_{d_j}}; \mathbb{P} : \Omega_{d_j} \mapsto [0, 1]; \sum_{w_i \in \Omega_{d_j}} p(w_i) = 1 \quad (2)$$

Eq. 1 indicates that any $d_j$ is a probability distribution over unique word-phrases $w_i$ from the sample space $\Omega_{d_j}$ where $w_i$s are lexicons in a vocabulary $\mathbf{V}$ (i.e., $d_j \in \mathcal{F}_{d_j}$). Similarly, we represent $u$ on the same space (i.e., for a given $d_j$, $u_j \in \mathcal{F}_{d_j}$) where we consider $u_j$ to be the expected summary of the reader for $d_j$ (in the evaluation setup, it will be the gold-reference summary). $s_{u_j}$ is also defined on the same space, and the extractive version can be considered as a subset of $d_j$ (i.e., for a given $(d_j, u_{jk})$, $s_{u_{jk}} \in \mathcal{F}_{d_j}$). We calculate the distributions as follows:

$$p(w_i|d_j) = \frac{count(w_i \in d_j)}{N_{d_j}} \quad (3)$$

$$r(w_i|s_{u_{jk}}) = \frac{\frac{count(w_i \in s_{u_{jk}})}{N_{s_{u_{jk}}}}}{p(w_i|d_j)} = \frac{p(w_i|s_{u_{jk}})}{p(w_i|d_j)} \quad (4)$$

$\{r(\bullet)$ indicates the ratio of $w_i\}$

$$\widehat{p}(w_i \in s_{u_{jk}}) = \frac{r(w_i|s_{u_{jk}})}{\sum\limits_{w_l \in \Omega} r(w_l|s_{u_{jk}})}$$

{Estimated probability distribution of summary}
$$\quad (5)$$

Continuing with the triangulation framework, the distance between a document $d$, the user profiles $u_i$ and $u_j$, and the corresponding user-specific summaries $s_{u_i}$ and $s_{u_j}$ generated by a model $M_{\boldsymbol{\theta},u}$ is calculated using Jenson-Shannon divergence (JSD) (Menéndez et al., 1997)[2] – a symmetrized version of the Kullback–Leibler divergence, which measures the similarity between two distributions. For calculating the divergence for the abstractive version of $s_{u_j}$ where we might encounter out-of-vocabulary (OOV) word-phrases (i.e., $p(w_i|d_j) = 0$ in Eq. 4, we propose a RoBERTa (Liu et al., 2019) embedding-based smoothing method in Section 3.3.

## 3.2 Deviation of a Summarization Model

Now that we have defined the triangulation space on which deviation can be computed, we propose a measure to calculate summary-level insensitivity of a model w.r.t subjectivity, called *Deviation-of-Summarization Model* ($Dev(s_{u_{ij}}|(d_i, u_{ij}))$) given reader-profiles (i.e., expected summaries or gold-references) and the document to be summarized.

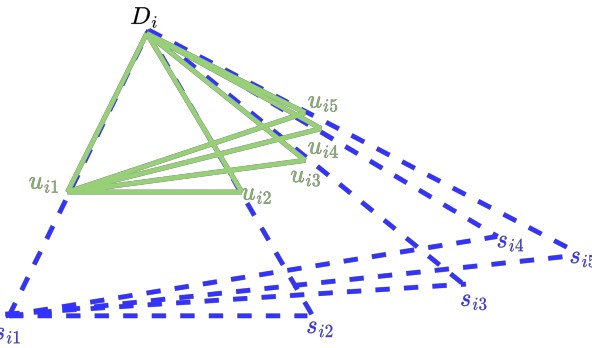

Figure 2: Summary-level deviation calculation of $s_{i1}$.

**Definition 4.** *Summary-level Deviation. Given a document $d_i$ and a reader-profile $u_{ij}$, the summary-level deviation of a model $M_{\boldsymbol{\theta},u}$ ($Dev(s_{u_{ij}}|(d_i, u_{ij}))$) is defined as the proportional divergence [3] between the summary $s_{u_{ij}}$ of $d_i$ that has been generated by $M_{\boldsymbol{\theta},u}$ for $u_{ij}$ from other reader-profile specific summaries of $d_i$ generated by $M_{\boldsymbol{\theta},u}$ w.r.t a corresponding divergence of $u_{ij}$ from other the reader-profiles (see Figure 2).*

We formulate $Dev(s_{u_{ij}}|(d_i, u_{ij}))$ as follows:

$$Dev(s_{u_{ij}}|(d_i, u_{ij})) = \frac{1}{|\mathbf{U}|} \sum_{k=1}^{|\mathbf{U}|} \frac{min(X_{ijk}, Y_{ijk})}{max(X_{ijk}, Y_{ijk})} \quad (6)$$

$$X_{ijk} = \frac{\exp(w(u_{ij}|u_{ik}))}{\sum\limits_{l=1}^{|\mathbf{U}|} \exp(w(u_{ij}|u_{il}))} \cdot JSD(u_{ij}||u_{ik}) \quad (7)$$

$$Y_{ijk} = \frac{\exp(w(s_{u_{ij}}|s_{u_{ik}}))}{\sum\limits_{l=1}^{|\mathbf{U}|} \exp(w(s_{u_{ij}}|s_{u_{il}}))} \cdot JSD(s_{u_{ij}}||s_{u_{ik}}) \quad (8)$$

$$w(u_{ij}|u_{ik}) = \frac{JSD(u_{ij}||u_{ik})}{JSD(u_{ij}||d_i)} \quad (9)$$

$$w(s_{uij}|s_{u_{ik}}) = \frac{JSD(s_{u_{ij}}||s_{u_{ik}})}{JSD(s_{u_{ij}}||d_i)} \quad (10)$$

Here $w(u_{ij}|u_{ik})$ and $w(s_{u_{ij}}|s_{u_{ik}})$ measure the relative divergence of $u_{ij}$ and the corresponding profile-specific summary $s_{u_{ij}}$ from the document $d_i$. A lower value of $Dev(s_{u_{ij}}|(d_i, u_{ij}))$ indicates that while reader-profiles are different, the generated summary $s_{u_{ij}}$ is very similar to other reader-specific summaries (or vice versa [4]), and hence, is not personalized at the summary-level.

---

[2]For two distributions P and Q:
$JSD(P\|Q) = \frac{1}{2}[D_{KL}(P\|M) + D_{KL}(Q\|M)]$
$M = \frac{(P+Q)}{2}$; where $D_{KL}$ is the KL divergence.

[3]In this paper, we have chosen Jenson-Shannon Divergence but both the triangulation space and the distance measure can be chosen to be something else as well.

[4]This ensures the strong condition of insensitivity.

### 3.3 Handling OOV

In Section 3.1, we discussed that smoothing is desirable for abstractive summaries. This is because there may be word-phrases in an abstractive summary, say $s_{u_{ij}}$, that are not present in the original document $d_i$ (i.e., Out-of-Vocabulary (OOV) word-phrases). This can seriously affect the divergence computations. To solve this, we propose a smoothing algorithm using contextual embeddings generated by RoBERTa (Liu et al., 2019). The central idea of the smoothing algorithm (see Appendix A for details) is to predict whether one or more of the OOV word-phrases present in $s_{u_{jk}}$ could be alternatives or augmentation to the word-phrases in the original document $d_j$. In the case of OOVs, we check how much $w_i \in d_j$ and $w_i^{OOV} \in s_{u_{jk}}$ are related by applying cosine similarity on their contextual word embeddings. A bias is added to capture the possibility of $w_i^{OOV}$ to be an unrelated addition in the summary. If the bias is higher than the closest match of $w_i^{OOV}$ in $d_j$, it indicates that $w_i^{OOV}$ is not an alternative/augmentation usage, and therefore, no smoothing is required. Else, smoothing is applied by taking the odds of $p(w_i^{OOV}|s_{u_{jk}})$ to $p(w_i^{OOV} \in d_j)$.

### 3.4 Effective Degree of Insensitivity

In Section 3.2, we defined the summary-level insensitivity of a model $M_{\boldsymbol{\theta},u}$. To determine the degree of insensitivity at a system level, we propose EGISES (*Effective deGree-of-InSEnsitivity w.r.t Subjectivity*). We formulate EGISES as follows:

$$\text{EGISES} = 1 - \frac{1}{|\mathbf{D}||\mathbf{U}|} \sum_{i=1}^{|\mathbf{D}|} \sum_{j=1}^{|\mathbf{U}|} Dev(s_{u_{ij}}|(d_i, u_{ij}))$$

(11)

A high value of EGISES indicates that the model implies that the model is insensitive to the reader's subjective preferences. As a result, a model with a high EGISES score would not be a good selection for use cases where we need personalized summaries.

## 4 Personalization of Existing Models

One of the key objectives of this paper is to analyze the degree of personalization of different state-of-the-art summarization models using EGISES. For this purpose, we chose ten different models, including abstractive and extractive summarizers. In this

section, we provide the framework for evaluating these models.

### 4.1 Evaluation Dataset

We use the test data in the PENS dataset[5] released by Microsoft Research to evaluate the models. It consists of news headlines together with news articles. The headlines could be considered extreme summaries of the corresponding news articles. The test set was created in two phases. In the first phase of data collection, 103 native English speakers were asked to browse through 1,000 news headlines and mark at least 50 pieces they were interested in. The headlines were randomly selected and arranged according to their first exposure time (this ensures the dataset has the reader's reading sequence captured as well). In the second phase, participants were asked to write their preferred headlines (i.e., gold references, and in our case, these become the reader-profile set **U**) for 200 different articles without knowing the original news title. These news articles were excluded from the first stage and were redundantly assigned to ensure that, on average, each news article has four gold-reference summaries (this makes this dataset suitable for testing insensitivity-to-subjectivity). The participants' click behaviors and more than 20,000 goal-reference personalized headlines of news articles were also collected, regarded as the expected summaries (Ao et al., 2021).

### 4.2 Models Studied

We studied ten off-the-shelf summarization models. We include five models from the PENS framework (Ao et al., 2021): PENS-NRMS Injection-Type 1 (or T1) and Injection-Type 2 (or T2), PENS-NAML T1, PENS-EBNR T1 & T2. We also include five more state-of-the-art models – BRIO (Liu et al., 2022), SimCLS (Liu and Liu, 2021), BigBird-Pegasus (Zaheer et al., 2020), ProphetNet (Qi et al., 2020), and T5-base (Orzhenovskii, 2021). Appendix B provides a brief description of each model. We select these models since they have been reported to be in the top 5 over the last four years on the CNN/Daily Mail news dataset, which is similar in content to the PENS dataset used for our evaluation.

---

[5]https://github.com/LLluoling/PENS-Personalized-News-Headline-Generation

### 4.3 Compared Accuracy Measures

To compare and correlate the leaderboard generated by `EGISES` with standard accuracy measures that have been reported to have sufficiently fair human-judgment correlation (and therefore, can be trusted), we select two ROUGE variants (RG-L (Lin and Och, 2004) and RG-SU4 (Lin, 2004)), BLEU-1 (Papineni et al., 2002), and METEOR (Banerjee and Lavie, 2005) (for a summary, see Appendix C.1). We use three standard correlation measures – Pearson's Correlation Coefficient ($r$), Spearman's $\rho$ Coefficient, and Kendall's $\tau$ Coefficient (see Appendix C.2).

## 5 Model Performance w.r.t `EGISES`

### 5.1 Experiment Design

There are three objectives regarding the state-of-the-art model evaluation: (i) leaderboard generation w.r.t `EGISES`, (ii) establishing that `EGISES` is a stable measure, and (iii) comparing the leaderboard with that of accuracy leaderboards. For the first two objectives, we conduct our experiments on ten random sample sets drawn from 100%, 80%, 60%, 40%, and 20% of the PENS test dataset. This is done to understand the bias and variance in the calculated `EGISES` scores, thereby measuring the stability of the measure. For the third objective, we use the correlation measures introduced in Section 4.3. For the PENS framework models, evaluation is direct since the models were designed to take user behavioral input in the way it is provided in the PENS dataset. However, for the other models, we need to appropriate the evaluation setup since they are not explicitly designed to be personalized. For these models, we add the gold reference summary of every reader as a title to the document that the models had to summarize, thereby creating multiple versions of the document corresponding to each reader. We then expect the model to take the injected titles as cues during summary generation, thereby inducing a personalization behavior. This injection also serves as a good baseline model.

### 5.2 Model Analysis & `EGISES` Stability

Our first observation (see Table 1) is that there is significant scope for improvement for personalized summarization considering the base models – the best PENS model is PENS-NAML (T1), which ranks sixth with an `EGISES` score of 0.8991 as compared to the best model (BigBird-Pegasus) that scores 0.4286. The induced personalization

in the generic models clearly helps them to significantly outperform the PENS models that were specifically designed for personalization. This also shows their ability to utilize the cue injection. At the same time, it shows that `EGISES` as a measure is clearly able to capture their ability to detect the injected cue and, hence, discriminate these models from the rest. Finally, we see no change of rank as we randomize our sample selection and average over ten draws for each set. The bias and variance fluctuation across sample size are very low, thereby showing the stability of `EGISES`.

### 5.3 Accuracy is Not Enough: Leaderboard Correlation Inconsistency

Table 2 shows that leaderboard correlation between that of `EGISES` and the other four accuracy measures w.r.t the three chosen correlation measures is inconsistent and inconclusive[6]. This supports our hypothesis that real-world datasets (such as PENS) have all possibilities of triangulations. Therefore, the theoretical proof that personalization and accuracy are unrelated (see Section 2.2) is also empirically established.

## 6 Reliability of `EGISES`

### 6.1 Meta Evaluation: Experiment Design

As a part of standard meta-evaluation of `EGISES`, collecting human-judgments for personalization can be practically infeasible. This is because to assess whether a model is insensitive or not, a human evaluator needs to go through a 3-step process: (i) give a judgment on the divergence between the expected summaries of the readers, (ii) a judgment on the divergence between the corresponding (personalized) summaries generated by any model, and then (iii) judge whether that divergence proportionally varies with expected summary divergences of the readers. This significantly differs from having the human judge provide a quality score on the model-generated summaries ($HJ$) and, therefore, requires significantly more resources and time to create such a dataset. One way to resolve this bottleneck is to utilize the (conditioned) transitivity property of Pearson's correlation (Langford et al., 2001) to establish a sufficiently high $r(HJ, \text{EGISES})$. Given that $r(HJ, Acc)$ is high (> 0.7) in most standard datasets such as CNN/DM and TAC-2008 (for RG-L) (Bhandari et al., 2020), and DUC-2001/2002 (for RG-L/SU-4) (Lin, 2004),

---

[6]Appendix 6 contains accuracy scores and ranking.

| | | PENS Test Dataset Sample Set (Random Selection) | | | | | | |
|---|---|---|---|---|---|---|---|---|
| **Ranking** | **Models** | **100%** | **80%** | **60%** | **40%** | **20%** | **Bias** | **Variance** |
| 1 | **BigBird-Pegasus** | **0.4286** | **0.4281** | **0.4208** | **0.4317** | **0.4265** | 0.0027 | 1.27E-05 |
| 2 | **SimCLS** | 0.5574 | 0.5583 | 0.5599 | 0.5576 | 0.5503 | 0.0025 | 1.11E-05 |
| 3 | **BRIO** | 0.6610 | 0.6599 | 0.6612 | 0.6604 | 0.6597 | 0.0005 | 3.70E-07 |
| 4 | **ProphetNet** | 0.8655 | 0.8650 | 0.8614 | 0.8661 | 0.8624 | 0.0017 | 3.42E-06 |
| 5 | **T5 (Base)** | 0.8817 | 0.8821 | 0.8844 | 0.8773 | 0.8872 | 0.0026 | 1.07E-05 |
| 6 | **PENS-NAML T1** | 0.8991 | 0.8999 | 0.8963 | 0.8998 | 0.9004 | 0.0016 | 2.17E-06 |
| 7 | **PENS-NRMS T1** | 0.9164 | 0.9166 | 0.9167 | 0.9142 | 0.9125 | 0.0011 | 2.85E-06 |
| 8 | **PENS-EBNR T1** | 0.9531 | 0.9536 | 0.9535 | 0.9543 | 0.9500 | 0.0011 | 1.79E-06 |
| 9 | **PENS-EBNR T2** | 0.9951 | 0.9947 | 0.9955 | 0.9957 | 0.9945 | 0.0004 | 1.97E-07 |
| 10 | **PENS-NRMS T2** | 0.9971 | 0.9975 | 0.9976 | 0.9973 | 0.9956 | 0.0005 | 5.33E-07 |

Table 1: Degree-of-personalization (Insensitivity-to-subjectivity) benchmark using EGISES on ten SOTA summarization models (lower score is better).

| | **Correlation Measure** | **RG-L** | **RG-SU4** | **BLEU** | **METEOR** |
|---|---|---|---|---|---|
| **EGISES** | Pearson $r$ | -0.3095 | -0.7879 | 0.1272 | -0.8814 |
| | Spearman $\rho$ | 0.0666 | -0.5555 | 0.2444 | -0.51111 |
| | Kendall $\tau$ | 0.1757 | -0.6848 | 0.2969 | -0.7333 |

Table 2: Personalization (using EGISES) vs. Accuracy based Leaderboard Rank Agreement Inconsistency.

one can therefore conclude high $r(HJ, \text{EGISES})$ if the condition below holds:

$$r(HJ, Acc)^2 + r(Dev_{Acc}, Dev_{\text{EGISES}})^2 > 1 \\ \implies r(HJ, \text{EGISES}) > 0 \quad (12)$$

$r(Dev_{Acc}, Dev_{\text{EGISES}})$ (calculated as per Figure 3) denotes the agreement in the degree-of-personalization ranking if we replace the proposed divergence measure ($Dev_{\text{EGISES}}$) introduced in Section 3.2 with that of standard accuracy measures that are used to measure divergence, but between model-generated summaries and reference-summaries. Therefore, according to Eq. 12, $r(Dev_{Acc}, Dev_{\text{EGISES}}) > 0.3$ would be sufficient to establish required $r(HJ, \text{EGISES})$.

## 6.2 Meta Evaluation: EGISES Reliability

We generate a degree-of-personalization leaderboard based on deviation calculation using all the selected standard accuracy measures (i.e., RG-L/SU-4, BLEU, METEOR). We can observe from Table 3 that in all cases, the Pearson correlation is $> 0.4$, with RG-L having a very high correlation of 0.8954. This result helps us to conclude that we can claim sufficiently high $r(HJ, \text{EGISES})$.

## 7 Personalized Accuracy

In this paper, we primarily focus on proving theoretically and establishing empirically that personalization is a separate attribute from accuracy (and therefore, the leaderboards of both do **not** correlate). Having said that, in this section, we show

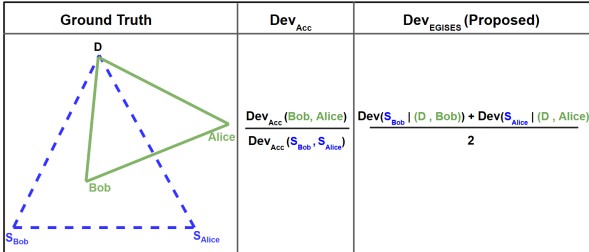

Figure 3: Correlating deviaton w.r.t a chosen accuracy measure $Dev_{Acc}$ and proposed $DINS_{\text{EGISES}}$.

that even when the primary objective of an evaluator is to measure the accuracy of a model, one should also take into account the ability of personalization of the model to get a realistic accuracy judgment. We call such an accuracy measure as $P - Acc$. However, we would like to emphasize that the key objective of $P - Acc$ is not to improve accuracy ranking but rather to achieve a more reliable accuracy **value** (or score) so that we can benchmark our personalized SOTA summarization models better and understand the scope of further improvement. Therefore, it is rather desirable that $P - Acc$ should have a high correlation with corresponding $Acc$. Considering this objective, this section aims to establish that EGISEScan be a good plugin for $P - Acc$.

### 7.1 Formulation

To design $P\text{-}Acc$ w.r.t a given accuracy measure $Acc$ and score $Score_{Acc}(M_{\boldsymbol{\theta},u})$, we can think of the model ($M_{\boldsymbol{\theta},u}$) performance as a vector in

| | Correlation Measure | $Dev_{RG-L}$ | $Dev_{RG-SU4}$ | $Dev_{BLEU}$ | $Dev_{METEOR}$ |
|---|---|---|---|---|---|
| $Dev_{\text{EGISES}}$ | Pearson $r$ | 0.8954 | 0.5187 | 0.4209 | 0.5639 |

Table 3: Analysis of personalization leaderboard Pearson correlation between $Dev_{Acc}$ and $Dev_{\text{EGISES}}$.

| Corr. | P-RG-L vs. RG-L | P-RG-SU4 vs. RG-SU4 | P-BLEU vs. BLEU | P-METEOR vs. METEOR |
|---|---|---|---|---|
| Pearson $r$ | 0.9707 | 0.9957 | 0.9795 | 0.9918 |
| Spearman $\rho$ | 0.6888 | 0.8090 | 0.9111 | 0.9555 |
| Kendall $\tau$ | 0.8182 | 0.9118 | 0.9636 | 0.9878 |

Table 4: Effect of $P\text{-}Accuracy$ on $Accuracy$ Leaderboard (higher is better)

$\mathbb{R}^2$: $\begin{bmatrix} Score_{Acc}(M_{\boldsymbol{\theta},u}) \\ \texttt{EGISES}(M_{\boldsymbol{\theta},u}) \end{bmatrix}$. Now, we can generalize $Score_{Acc}(M_{\boldsymbol{\theta},u})$ as a multiplication of the score with its associated unit-scale ($Unit_{Acc}$). In standard cases, $Unit_{Acc} = 1$. We propose that to understand the accuracy of personalized summarization models, the model's original accuracy score should be penalized. In other words, $Unit_{Acc}$ should be penalized in proportion to the $\texttt{EGISES}$ score as follows,

$$Unit_{P\text{-}Acc} = 1 - [\alpha \cdot (\frac{f_{sig}(\beta \cdot \texttt{EGISES}(M_{\boldsymbol{\theta},u}))}{Score_{Acc}(M_{\boldsymbol{\theta},u})})] \tag{13}$$

Here, $\beta \in (0, 1]$ is the *personalization-coefficient* that controls how much importance one can expect to give to the personalization dimension of the vector (i.e., the squashing of the corresponding personalization basis vector)[7]. $\alpha \in [0, 1]$ is the *compensation-coefficient* and regulates the final penalty that needs to be applied. $f_{sig}$ is a sigmoid function to prevent the original accuracy score from getting overly dampened. As a result, we can now formulate $P\text{-}Acc$ as,

$$P\text{-}Acc(M_{\boldsymbol{\theta},u}) = Score_{Acc}(M_{\boldsymbol{\theta},u}) \cdot Unit_{P\text{-}Acc} \tag{14}$$

We take $\beta = 1$ to understand the extreme case effect of $\texttt{EGISES}$ on the original accuracy leaderboard and $\alpha = 0.5$ so as not to over-penalize a fairly accurate model for poor personalization. It is up to the evaluator to decide how much personalization is to be emphasized (controlled by $\beta$ value) during accuracy evaluation and what percentage of the overall penalty (due to lack of personalization) (controlled by $\alpha$) should be injected into the original accuracy score value.

### 7.2 Meta Evaluation: P-Acc Leaderboard Correlation

To analyze how stable $P\text{-}Acc$ is when compared to changes in the corresponding accuracy measure,

---

[7] $\beta = 0$ would result in $Unit_{P\text{-}Acc} = Unit_{Acc}$.

we perform a correlation analysis and find that the Kendall $\tau$ coefficient is highly positive (lowest of 0.8182 for RG-L and maximum of 0.9878 for METEOR) (see Table 4 for details). This means incorporating personalization into the accuracy measures will not adversely affect a model's expected accuracy. However, $P\text{-}Acc$ can give us a more realistic understanding of how accurate a personalized summarization model is.

## 8 Related Work

Personalization in summarization models can be broadly categorized into two types – *iterative human feedback based models* (IHF models), and *user-profile based models*. In IHF-based models, the reader keeps giving feedback on the summary iteratively till the reader is satisfied with the model-generated summary. Ghodratnama et al. (2020b) proposed a personalized summarization approach in which Exdos (Ghodratnama et al., 2020a) is used as a base model for extractive summarization model to rank sentences of news body, and then concepts are extracted and shown to readers for their feedback. The system uses this feedback to iteratively generate a summary that includes the most important concepts till no further negative feedback. However, the evaluation metric used was ROUGE variants as accuracy metrics, and as established in this paper, cannot be used for measuring personalization. They also measured the change in ROUGE as the iterations increased, but that still is not a measure of subjective deviation. On the other hand, user-profile-based models such as those that were designed using the PENS framework (Ao et al., 2021), which we studied extensively, need significant improvement in personalization. These models, too, were evaluated using ROUGE variants w.r.t accuracy. In the area of personalized recommendation and search, we can find Jaccard Index-based measures, order edit distance-based measures (Hannak et al., 2013), and the popular nDCG (normalized Discounted Cumulative Gain)-

based measures (Matthijs and Radlinski, 2011). However, these are not directly applicable to text summarization and are also extrinsic, relying on human feedback (clicks, likes, etc.), and therefore, cannot be automatic intrinsic measures.

## 9 Conclusion

In this paper, we establish theoretically and empirically that accuracy measures are unsuitable to evaluate the degree-of-personalization of summarization models, specifically when saliency is fairly subjective. We introduce a new measure, called EGISES, and show that EGISES is both stable w.r.t bias and variance and reliable w.r.t human-judgment correlation when we analyzed the degree-of-personalization of ten summarization models. As an extension, EGISES needs to be appropriated for capturing the sensitivity of models to the time-variant evolving interests of readers.

## Limitations

As discussed in Section 6.1, creating a dataset for direct meta-evaluation of EGISES is crucial to have a quantitative understanding of the correlation between human judgment and the proposed measure. At present, we have only been able to establish that there is a sufficiently high correlation. We are currently in the process of opening up an online survey specifically for the creation of this dataset. Also, currently, we are not in a position to conclude what kind of metric space EGISES should be defined (in this paper, we took a probability space) and what distance metric for measuring deviation would yield the best human-judgment correlation (we used Jenson-Shannon Divergence and tried to understand how that correlated if replaced by standard accuracy measures but as deviation measures). Finally, we do not have conclusive results of the effect of different contextual embeddings from other SOTA LLMs (specifically GPT-X models) on the overall leaderboard and human-judgment correlation. Thorough ablation studies are required.

## Ethics Statement

We believe there needs to be a formal framework within which both theoretical and empirical analysis of the personalization capabilities of contemporary large language models (LLMs) can be evaluated. Personalization is not only an important aspect of "intelligence", but it also elucidates the LLMs' capabilities of discerning what is subjectively valued and, more importantly, what is not. We would like to highly encourage fellow researchers to do a serious review of this work and build on the proposed EGISES framework. We would also like to declare that we used the PENS dataset prepared and released by Microsoft Research and did not involve any human subject for any part of the evaluation or meta-evaluation.

## Acknowledgements

This work could not have been completed in time without all the accuracy results generated by Isha Motiyani (202218015@daiict.ac.in), Nidhi Somaiya (202218060@daiict.ac.in), and Radhika Singhal (202218062@daiict.ac.in) who are research assistants at KDM Lab, DA-IICT, India.

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

## A  Out-of-Vocabulary Smoothing

While generating the distribution of the summary, there can be $w_i^{OOV} \in s_{u_{jk}}$, that are present in the summary but not in the document itself, in that case rather assigning 0 probability assuming that the word is not part of the document, we proposed an algorithm to handle such cases. We proposed Out-of-Vocabulary Smoothing algorithm that calculates the probability of having an alternative or augmentation of the $w_i^{OOV}$ word in the document. To explain the algorithm with an example, consider a toy document: "*The red cat is on the red tall table.*" which after preprocessing would be the bow: {red, cat, red, tall, table}. Let's say that the reader's expected summary (gold-reference) is: "*The cat is on table*" whose bow is: {cat, table}. Now, given the document and the reader's profile an abstractive model generates a summary "*The red cat is on the desk*", desk being the $w_i^{OOV}$, and whose bow is: {red, cat, desk}. As per table 5, we can see that desk can be considered as an alternative to *table*, and therefore gets a smoothened probability mass of 0.3805 by doing a sum of all softmax ($\sigma$) similarity score of words in the document. Intuitively, softmax similarly score acts as the probability of that word being alternative or augmentation of the $w_i^{OOV} \in s_{u_{jk}}$. It's possible that the model generates a completely new word, i.e., unrelated to any of the words in the document, which means the probability of that word in the document should be 0. In our algorithm, bias gets the highest similarity score among all scores in that case because bias is calculated from the highest similarity score of words in the document, and the highest similarity word in the document itself gets a low similarity score.

## B  Models Details

We briefly introduce the SOTA summarization models that were analyzed to understand their degree-of-personalization below:

1. **PENS-NRMS Injection-Type 1**: The PENS framework (Ao et al., 2021) takes user embedding as input along with the news article to generate a personalized summary for that user. To generate user embedding NRMS

---

[7]In our implementation, we used the *all-distilroberta-v1* model from Huggingface. This model maps sentences paragraphs to a 768-dimensional dense vector space and has 6 layers, 768 dimensions and 12 heads, and 82M parameters.

| $w_i^{OOV} \in s_u$ | $w_i \in d$ | $f_{cos}(\mathbf{e}_{desk}, \mathbf{e}_{w_i})$ | $\sigma(f_{cos}(\mathbf{e}_{desk}, \mathbf{e}_{w_i}))$ | $OOV[desk]$ |
|---|---|---|---|---|
| | red | 0.537 | 0.2 | |
| | cat | 0.405 | 0.175 | |
| desk | tall | 0.613 | 0.216 | **0.3805** |
| | table | **0.892** | 0.285 | |
| | bias[desk] | $1 - \sqrt{0.89} < 0.892$ | 0.124 | |

Table 5: OOV handling example

---

**Algorithm 1** Out-of-Vocabulary Smoothing

> **Input**: $d_j, s_{u_{jk}}$
> **Output**: $OOV$ {Estimated probabilities of OOV word-phrases as alternatives/augmentation to document phrases}
> **procedure** GETEMBEDDING($d_j$, RoBERTa)
>     **for** $\{w_i^{OOV} \in s_{u_{jk}} | w_i^{OOV} \notin d_j\}$ **do**
>         $\mathbf{e}_{w_i^{OOV}} \leftarrow$ GetEmbedding($\mathrm{w}_i^{OOV}, RoBERTa$)
>
>     $max_{sim}[i] \leftarrow \max_l \{f_{cos}(\mathbf{e}_{w_i^{OOV}}, \{\mathbf{e}_{w_l} | w_l \in d_j\})\}$
>
>     $bias[i] \leftarrow 1 - \sqrt{score_{sim}[i]}$
>
>     **if** $bias[i] > score_{sim}[i]$ **then**
>         $OOV[i] \leftarrow 0$ {$w_i^{OOV}$ is **not** an alternative/augmentation usage (i.e. probability = 0, or no smoothing required)}
>     **else**
>
>     $sum_{sim}[i] \leftarrow \sum_l \sigma(f_{cos}(\mathbf{e}_{w_i^{OOV}}, \{\mathbf{e}_{w_l} | w_l \in d_j\}))$
>
> {$\sigma : Softmax(\bullet)$}
> $OOV[i] \leftarrow \frac{count(w_i^{OOV})/N_{s_{u_{jk}}}}{sum_{sim}[i]}$ {$N_{s_{u_{jk}}}$: total number of word-phrases in $s_{u_{jk}}$.}
>     **return** $OOV$

(Neural News Recommendation with Multi-Head Self-Attention) (Wu et al., 2019b) is used. It includes a news encoder that utilizes multi-head self-attentions to understand news titles. The user encoder learns user representations based on their browsing history and uses multi-head self-attention to capture connections between news articles. Additive attention is added to learning the news and user representations more effectively by selecting important words and articles. Here, Injection-Type 1 indicates that NRMS user embedding is injected into PENS by initializing the de-coder's hidden state of the headline generator, which will influence the summary generation.

2. **PENS-NRMS Injection-Type 2**: To generate a personalized summary, NRMS user embedding is injected into attention values (Injection-Type 2) of PENS that helps to personalize attentive values of words in the news body.

3. **PENS-NAML Injection-Type 1**: NAML (Neural News Recommendation with Attentive Multi-View Learning) (Wu et al., 2019a) incorporates a news encoder that utilizes a multi-view (i.e., titles, bodies, and topic categories) attention model to generate comprehensive news representations. The user encoder is designed to learn user representations based on their interactions with browsed news. It also allows the selection of highly informative news during the user representation learning process. This user embedding is injected into the PENS model using Type-1 for personalization.

4. **PENS-EBNR Injection-Type 1**: EBNR (Embedding-based News Recommendation for Millions of Users) (Okura et al., 2017) proposes a method for user representations by using an RNN model that takes browsing histories as input sequences. This user embedding is injected using Type 1 into the PENS model for personalization.

5. **PENS-EBNR Injection-Type 2**: This personalized model injects EBNR user embedding into PENS using type-2.

6. **BRIO**: Instead of a traditional MLE-based training approach, BRIO (Liu et al., 2022) assumes a non-deterministic training paradigm that assigns probability mass to different candidate summaries according to their quality, thereby helping it to better distinguish

between high-quality and low-quality summaries.

7. **SimCLS**: SimCLS (A Simple Framework for Contrastive Learning of Abstractive Summarization) (Liu and Liu, 2021) uses a two-stage training procedure. In the first stage, a Seq2Seq model (BART (Lewis et al., 2020)) is trained to generate candidate summaries with MLE loss. Next, the evaluation model, initiated with RoBERTa is trained to rank the generated candidates with contrastive learning.

8. **BigBird-Pegasus**: BigBird (Zaheer et al., 2020) is an extension of Transformer based models designed specifically for processing longer sequences. It utilizes sparse attention, global attention, and random attention mechanisms to approximate full attention. This enables BigBird to handle longer contexts more efficiently and, therefore, can be suitable for summarization.

9. **ProphetNet**: ProphetNet (Qi et al., 2020) is a sequence-to-sequence pre-trained model that employs n-gram prediction using the n-stream self-attention mechanism. ProphetNet optimizes n-step ahead prediction by simultaneously predicting the next n tokens based on previous context tokens, thus preventing overfitting on local correlations.

10. **T5**: T5 (Text-To-Text Transfer Transformer) is based on the Transformer-based Encoder-Decoder architecture that operates on the principle of the unified text-to-text task for any NLP problem, including summarization. Some recent analysis on the performance of T5 on summarization task can be found in (Tawmo et al., 2022; Ramesh et al., 2022; Etemad et al., 2021).

## C  Accuracy and Performance

### C.1  Accuracy Measures Compared

1. **RG-L**: ROUGE-L (Recall-Oriented Understudy for Gisting Evaluation) (Lin and Och, 2004) calculates the longest common subsequence between the generated summary and the reference summary and then measures the precision, recall, and F1 score based on this comparison.

2. **RG-SU4**: In addition to capturing unigram, bigram, and trigram matches, ROUGE-SU4 (Lin, 2004) also considers skip-bigram matches, which allow for gaps of certain words between the matched n-grams, thereby also considering non-contiguous n-gram matches.

3. **BLEU**: BLEU (Bilingual Evaluation Understudy) (Papineni et al., 2002) is a popular evaluation metric that measures the precision of n-gram matches between the model-generated summaries and the reference summaries. BLEU computes a modified precision score for various n-gram lengths and then combines them using a geometric mean.

4. **METEOR**: METEOR (Metric for Evaluation of Translation with Explicit ORdering) (Banerjee and Lavie, 2005) matches unigrams based on surface forms, stemmed forms, and meanings and then calculates score using a combination of precision, recall, and the order-alignment of the matched words w.r.t reference summary.

### C.2  Correlation Measures

1. **Pearson's Correlation Coefficient** ($r$):

$$r = \frac{\sum_{i=1}^{n}(x_i - \overline{x})(y_i - \overline{y})}{\sqrt{\sum_{i=1}^{n}(x_i - \overline{x})^2 \sum_{i=1}^{n}(y_i - \overline{y})^2}}$$

where $\overline{x}, \overline{y}$ are the means of the variables $x_i$ and $y_i$ ; $n$ = the number of samples.

2. **Spearman's $\rho$ Coefficient**:

$$\rho = 1 - \frac{6 \sum d_i^2}{n(n^2 - 1)}$$

where $d$ = the pairwise distances of the ranks of the variables $x_i$ and $y_i$ ; $n$ = the number of samples.

3. **Kendall's $\tau$ Coefficient**:

$$\tau = \frac{c - d}{c + d} = \frac{S}{\binom{n}{2}} = \frac{2S}{n(n - 1)}$$

where, $c$ = the number of concordant pairs; $d$ = the number of discordant pairs.

| | | | Leaderboard (PENS test dataset) | | |
|---|---|---|---|---|---|
| **Models** | EGISES | **RG-L** | **RG-SU4** | **BLEU** | **METEOR** |
| **BigBird-Pegasus** | 1 | 1 | 1 | 1 | 3 |
| **SimCLS** | 2 | 7 | 2 | 7 | 1 |
| **BRIO** | 3 | 8 | 6 | 8 | 2 |
| **ProphetNet** | 4 | 9 | 3 | 9 | 4 |
| **T5 (Base)** | 5 | 10 | 7 | 10 | 5 |
| **PENS-NAML T1** | 6 | 4 | 5 | 6 | 10 |
| **PENS-NRMS T1** | 7 | 2 | 9 | 5 | 9 |
| **PENS-EBNR T1** | 8 | 6 | 10 | 2 | 7 |
| **PENS-EBNR T2** | 9 | 3 | 4 | 4 | 8 |
| **PENS-NRMS T2** | 10 | 5 | 8 | 3 | 6 |

Table 6: Personalization vs. Accuracy: Leaderboard rank disagreement

## C.3 Accuracy Leaderboard

In table 6, each column contains the ranking of the summarization model as per that specified measure. The table is sorted by the ranking of EGISESscore. The ranking of a model is inconsistent across different measures due to that the correlation between EGISESand the other four accuracy measures is low.