# OpenReview forum: "Accuracy is not enough: Evaluating Personalization in Summarizers"
_EMNLP/2023/Conference — EMNLP 2023 Findings_

### Official Review · Reviewer_GHAL · 2023-08-04

**Soundness:** 3

**Excitement:**

3: Ambivalent: It has merits (e.g., it reports state-of-the-art results, the idea is nice), but there are key weaknesses (e.g., it describes incremental work), and it can significantly benefit from another round of revision. However, I won't object to accepting it if my co-reviewers champion it.

**Paper Topic And Main Contributions:**

This paper focuses on the personalization evaluation in the summarization task. The authors propose EGISES to measure the degree of personalization.

**Questions For The Authors:**

See above.

**Reasons To Accept:**

* The paper is well-written and structured.

* The paper contains several theoretical proofs.

**Reasons To Reject:**

The primary concerns include,

* The necessity of evaluating the degree of personalization is not clear to me.

  - According to this paper, I only found three previous research that did personalized summarizers. And all of them utilize the current common metrics to measure performance. It seems these metrics are enough for this task.

  - Let's assume the new evaluation metric is necessary. When we have the pairs of user profiles (such as user-expected summaries) and generated summaries for each user, why can we not use the average and variance of current metrics (such as Rouge) to show the degree of personalization? The average presents the performance of the summarizer to generate high-quality summaries, and variance can represent the performance of generating summaries close to each user. It is much easier to evaluate based on current metrics than new ones.

* The new proposed metric is only tested on a single dataset.

* There is no human judgment for this new metric. I notice the authors said, in Limitations, they are trying for the human evaluation. I think it is better to accept the next version with human judgment results.

* The metric is high time-cost due to the eight Jenson-Shannon Divergence calculations.

Besides, the details of $rot()$ were missed in Line 184.

**Reproducibility:**

3: Could reproduce the results with some difficulty. The settings of parameters are underspecified or subjectively determined; the training/evaluation data are not widely available.

**Reviewer Confidence:**

4: Quite sure. I tried to check the important points carefully. It's unlikely, though conceivable, that I missed something that should affect my ratings.

**Typos Grammar Style And Presentation Improvements:**

* Line 177, it should be the distance between two spaces rather than RATIO.

---

> ### Author Rebuttal · Authors · 2023-08-28
>
> $\textbf{Need for separate personalization measure:}$
> We have proven mathematically that deviation varies while keeping accuracy constant. Using the same proof technique (i.e., rotation operator), we can also argue (mathematically) that subjective accuracy can vary (i.e., can have a total ordering) while keeping deviation constant. Since deviation is at the core of personalization evaluation, accuracy variance as a measure, therefore, will not be able to capture the degree of personalisation. Also, all the standard accuracy measures do not weigh in user-specific word preferences. This can result in a high variance in accuracy without any discrimination of core-topic match vs. peripheral-topic match between generated summaries and reference summaries, which is at the heart of personalization. Finally, we observed empirically that the accuracy leaderboard has no conclusive correlation with that of EGISES leaderboard at system-level and also at $\textit{summary-level}$, thereby supporting the fact that in real-world datasets such as PENS, accuracy variance is not good enough.
>
> It is noteworthy that if the same accuracy measures (such as ROUGE) are plugged into EGISES or a simplified EGISES (see Table 3) instead of JSD, then the personalization leaderboard correlation is acceptable (with RG-L having a high correlation of 0.89). In other words, the issue is not with the measures themselves (which could be used as alternatives in the proposed EGISES framework) but rather with using these measures for accuracy evaluation and expecting that we can evaluate personalization implicitly.
>
> $\textbf{Clarification on the use of PENS dataset only:}$
> The concern is valid. However, one must keep in mind that the five SOTA personalized summarization models we evaluated do user behavior modeling by embedding the temporal sequence in which a user interacts with documents, i.e., clicking, reading, and writing a reference summary (i.e., expected summary). For this, we need a dataset that should contain the same document to have multiple users' reference summaries together $\textbf{with the user-interaction sequence timestamp}$. So far, we have not encountered any dataset that satisfies the criteria except the PENS dataset. We would love to include any similar dataset in our future studies whenever we find one.
>
> $\textbf{Clarification on lack of human judgment:}$
> The primary objective of accepting any newly proposed measure as reliable is to show sufficient positive human judgment correlation. This has already been established in the paper in Section 6.2 using equation 12 (i.e., the correlation transitivity property; see Section 6.1), where already published human-correlation results on the news dataset (CNN/DM) are being used. We are currently understanding, quantitatively, how much this positive correlation is via a survey-based human judgment dataset creation and how much we can increase the positive correlation using different alternative distance measures within the EGISES framework.
>
> $\textbf{Survey setup and initial results:}$ We have collected 110 participant responses so far (61 female, 49 male). Each participant (undergraduate and graduate students studying computer science, mathematical science, electronic engineering, and humanities) is shown six pairs of summaries from the PENS dataset such that one pair consists of randomly selected users' reference summaries, and the other five are corresponding model-generated summaries from five different models. So, with two consecutive participants who are shown the same pair of users' reference summaries, we are able to cover all the ten models we have evaluated along with 55 documents. The participants are not told that the pairs are summaries to avoid any prior bias. Instead, the pairs are shown as plain texts, and they are asked to provide a similarity rating between 1 (low) and 6 (very high) for every pair. This way, we can collect their unbiased judgment of how close two (subjective) user reference summaries are and how close their corresponding model-generated summaries are. We have so far observed a sufficiently high model-leaderboard correlation between EGISES and human judgment ($\textbf{0.6197}$ Pearson's $r$; $\textbf{0.5111}$ Kendall $\tau$; $\textbf{0.7333}$ Spearman $\rho$).
>
> $\textbf{Computational complexity of EGISES:}$ The computational complexity can be a legitimate concern if it has an unreasonable wall-clock runtime. We observed an average runtime of 2:37 minutes over an evaluation dataset of 3840 documents (each containing four user reference summaries and corresponding model-generated summaries). So, evaluating all 10 models will not take more than 25 minutes, which is reasonable for an offline procedure. $\textbf{NOTE: System specifications}:$ Machine architecture: x86\_64; CPU: Intel(R) Xeon(R) Silver 4216 CPU @ 2.10GHz; CPU Cores: 16; Thread(s) per core: 2. Also, as shown below, it can be proved that EGISES will not be computationally more costly (asymptotically) when compared to any other alternative and more simplistic metric.
> $\textbf{Complexity analysis:}$ The time complexity of each summary-level deviation $Dev(s_{u_{ij}} |(d_i, u_{ij}))$: $|U| \times O(4 \times t_{JSD}) = O(|U| \cdot t_{JSD})$, where $|U|$ is the total number of users in the evaluation dataset and $t_{JSD}$ is the time-complexity of computing JSD (there are 4 JSD computations and not 8, please refer Equations 9 and 10). The worst-case $t_{JSD} = O(l_s+l_u)$, where $l_s, l_u$ are lengths of $s, u$. Since any degree-of-personalization metric will have to be comparison-based, the lower bound will be $\Omega(|U| \cdot \sigma(\bullet,\bullet))$. If $\sigma$ is $ROUGE$ (i.e., we use $ROUGE$ as a deviation measure instead of an accuracy measure), then applying suffix-tree-based matching, we can obtain a worst-case complexity of $O(l_s+l_u)$. So, even with a relatively simple measure, the best we could have achieved remains the same asymptotically - i.e., $O(|U| \cdot (l_s+l_u))$.

---

### Official Review · Reviewer_QpF4 · 2023-08-06

**Soundness:** 3

**Excitement:**

4: Strong: This paper deepens the understanding of some phenomenon or lowers the barriers to an existing research direction.

**Paper Topic And Main Contributions:**

The paper presents theoretical and empirical evidence that accuracy measures are inadequate for assessing the degree of personalization in summarization models, especially when saliency is subjective. To address this limitation, the authors propose a new measure called EGISES. Their analysis of ten state-of-the-art summarization models demonstrates that EGISES is stable concerning bias and variance and exhibits a reliable correlation with human judgment in measuring the degree of personalization.

**Questions For The Authors:**

-  Line 176: the definition of degree-of-personalization and Figure caption seems contradictory. Assuming figure 1 is correct, shouldn’t the ratio be inverted?

-  Why didn't you include BERTScore in the evaluation?

**Reasons To Accept:**

- Personalization in summarization is an important task and needs more attention regarding evaluation. Therefore, this paper tackles a very important problem.

- The paper provides theoretical and empirical evidence that EGISES can model the personalization of a summarization model.

- The paper is nicely written, it was a pleasure to read.

**Reasons To Reject:**

- Section 7.2 is hard to understand for me. What were the alpha and beta values you used for Table 6? I could not find any details on that. Table 6 results will vary based on alpha and beta, and it is important to study that sensitivity.

- Table 6 also shows that P-Acc vs. Acc correlations are very high for Pearson and Kendall for all Acc metrics. I find it a little puzzling. The authors claim that:  "This means that incorporating personalization into the accuracy measures will not have any adverse effect on the expected accuracy of a model. However, P-Acc will give us a more realistic understanding of how accurate a personalized summarization model is." I'm not sure if I see the point here. Why a high correlation is a good thing here? A high correlation between P-Acc vs. Acc means they essentially give us the same conclusion. Then, how is P-Acc helping? Am I missing something here?

- Equation 13 is not clearly described. What is "fsig" here? What is the overall justification of this equation?

**Reproducibility:**

4: Could mostly reproduce the results, but there may be some variation because of sample variance or minor variations in their interpretation of the protocol or method.

**Reviewer Confidence:**

4: Quite sure. I tried to check the important points carefully. It's unlikely, though conceivable, that I missed something that should affect my ratings.

---

> ### Author Rebuttal · Authors · 2023-08-28
>
> $\textbf{Clarification on $P\text{-}Acc$ correlation with $Acc$:}$ The paper primarily focuses on proving theoretically and establishing empirically that personalization is a separate aspect from accuracy (and therefore, the leaderboards of both do $\textbf{not}$ correlate). Having said that, in section 7, we show that even if the primary objective of an evaluator is to measure the accuracy of a model, one should take into account the ability of personalization of the model to get a $\underline{realistic}$ accuracy judgment. We call such an accuracy measure as $P-Acc$. However, the key objective of $P-Acc$ is not to improve accuracy ranking but rather to achieve a more reliable accuracy $\textbf{value}$ so that we can benchmark our personalized SOTA summarization models better and understand the scope of further improvement. Therefore, it is rather desirable that $P-Acc$ should have high correlation with corresponding $Acc$. Considering this objective, we have successfully established that EGISES can be a good plugin for $P-Acc$. It is up to the evaluator to decide how much personalization is to be emphasized (controlled by $\beta$ value) during accuracy evaluation and what percentage of the overall penalty (due to lack of personalization) (controlled by $\alpha$ value) should be injected into the original accuracy score.
>
> $\textbf{Clarification on $f_{sig}$ and hyper-parameter setup:}$ $f_{sig}$ is a sigmoid function for squashing (i.e. reducing) the penalty score for higher EGISES values, thereby preventing the original accuracy score from getting dampened too much. In our experiment $\beta = 1$ since we wanted to understand the extreme case effect of EGISES on the original accuracy leaderboard, and $\alpha = 0.5$ (since we did not want to over-penalize a model for poor personalization that otherwise could be performing well in terms of accuracy alone). We will incorporate this detail in the camera-ready if accepted.
>
> $\textbf{Rationale for exclusion of BERTScore:}$ There were primarily three reasons to exclude BERTScore: (i) according to findings in Deutsch et al. [EMNLP, 2022], BERTScore was reported to have a significantly lower human correlation on RealSumm (news dataset based on CNN/DM where relevancy annotation was done using a lightweight version of PYRAMID (a highly rigorous manual summarization evaluation metric) in comparison to the metrics that we have chosen, (ii) the system ranking (i.e., model leaderboard) stability of BERTScore was found to be significantly lower, as was evident from the relatively large 95\% CI (Confidence Interval), (iii) according to the findings in Fabbri et al. [TACL, 2021], the inter-correlation of system leaderboard generated by BERTScore with that of RG-based metrics was shown to be very high on SummEval dataset (news dataset based on CNN/DM), as compared to other metrics such as METEOR and BLEU. Based on these results, we decided that contrasting the accuracy leaderboard w.r.t BERT with that of the personalization leaderboard w.r.t EGISES will be redundant. Hence, for lack of strong motivation to include BERTScore, we decided to abandon the experiment that otherwise would have been computationally very expensive.
>
> $\textbf{Deviation Ratio inconsistency in the formula:}$ The deviation ratio, described on page 2, is consistent in that the denominator will always be greater than or equal to the numerator because of a max/min normalization (see equation 6 for clarification). However, the total ordering property, as argued in the proof, and therefore the argument that deviation varies while accuracy is constant, is independent of what we keep in the numerator.

---

### Official Review · Reviewer_wqBM · 2023-08-12

**Typos Grammar Style And Presentation Improvements:** Formulas in Preliminaries should also…
**Soundness:** 4

**Excitement:**

4: Strong: This paper deepens the understanding of some phenomenon or lowers the barriers to an existing research direction.

**Paper Topic And Main Contributions:**

The authors propose a new metric, EGISES, to measure personalization in summary generation. The paper proves the theoretical feasibility of measuring personalization, and then proposes a personalization measurement method based on JSD divergence.

**Questions For The Authors:**

Please refer to the content of “Reasons To Reject”.

**Reasons To Accept:**

* The paper theoretically proves that EGISES can model personalization for the summarization task.
* The correlation analysis with the existing measures proves that P-acc differs from the existing measures.
* The paper is well written and organized.

**Reasons To Reject:**

* The paper only compares with the measures of the ROUGE series and lacks comparisons with other classic measures, such as BERTScore.
* Equation 9-10: EGISES needs to calculate the JSD divergence multiple times, and the computational complexity is relatively high.


**Reproducibility:**

4: Could mostly reproduce the results, but there may be some variation because of sample variance or minor variations in their interpretation of the protocol or method.

**Reviewer Confidence:**

4: Quite sure. I tried to check the important points carefully. It's unlikely, though conceivable, that I missed something that should affect my ratings.

---

> ### Author Rebuttal · Authors · 2023-08-28
>
> $\textbf{Rationale for Exclusion of BERTScore:}$ There were primarily three reasons to exclude BERTScore: (i) according to findings in Deutsch et. al. [EMNLP, 2022], BERTScore was reported to have a significantly lower human correlation on RealSumm (news dataset based on CNN/DM where relevancy annotation was done using a lightweight version of PYRAMID - a highly rigorous manual summarization evaluation metric) in comparison to the metrics that we have chosen, (ii) the system ranking (i.e., model leaderboard) stability of BERTScore was found to be significantly lower, as was evident from the relatively large 95\% CI (Confidence Interval), (iii) according to the findings in Fabbri et. al. [TACL, 2021], the inter-correlation of system leaderboard generated by BERTScore with that of RG-based metrics was shown to be very high on SummEval dataset (news dataset based on CNN/DM), as compared to other metrics such as METEOR and BLEU. Based on this result we decided that contrasting accuracy leaderboard w.r.t BERT with that of personalization leaderboard w.r.t EGISES will be redundant. Hence, for lack of strong motivation to include BERTScore, we decided to abandon the experiment that otherwise would have been computationally very expensive.
> Also, apart from ROUGE variants (RG-L, RG-SU4) we have included other non-ROUGE metrics like BLEU and METEOR.
>
> $\textbf{Computational Complexity of EGISES:}$ The computational complexity can be a legitimate concern if it has an unreasonable wall-clock runtime. We observed an average runtime of 2:37 minutes over an evaluation dataset of 3840 documents (each document containing 4 user reference summaries and corresponding model generated summaries). So, evaluating all 10 models will not take more than 25 minutes, which is reasonable for an offline procedure. $\textbf{NOTE: System specifications}:$ Machine architecture: x86\_64; CPU: Intel(R) Xeon(R) Silver 4216 CPU @ 2.10GHz; CPU Cores: 16; Thread(s) per core: 2. Also, as shown below, it can be proved that EGISES will not be computationally more costly (asymptotically) when compared to any other alternative and more simplistic metric.
> $\textbf{Complexity Analysis:}$ The time complexity of each summary-level Deviation $Dev(s_{u_{ij}} |(d_i, u_{ij}))$: $|U| \times O(4 \times t_{JSD}) = O(|U| \cdot t_{JSD})$, where $|U|$ is the total number of users in the evaluation dataset and $t_{JSD}$ is the time-complexity of computing JSD. The worst-case $t_{JSD} = O(l_s+l_u)$, where $l_s, l_u$ are lengths of $s, u$. Since any degree-of-personalization metric will have to comparison-based, therefore the lower-bound will be $\Omega(|U| \cdot \sigma(\bullet,\bullet))$. If $\sigma$ is $ROUGE$ (i.e., we use $ROUGE$ as a deviation measure instead of accuracy measure) then applying suffix-tree based matching we can obtain a worst-case complexity of $O(l_s+l_u)$. So, even with a relatively simple measure the best we could have achieved remains the same asymptotically - i.e., $O(|U| \cdot (l_s+l_u))$.

---

### Meta-Review · Area_Chair_ogQ7 · 2023-09-17

**Recommendation:** 3

**Metareview:**

The paper introduces EGISES, a novel metric for evaluating personalization in summarization. Reviewers acknowledge the paper's theoretical contributions and its importance in addressing personalization in summarization, praising its clarity and organization. However, concerns include limited comparisons with classic metrics like BERTScore, computational complexity due to multiple JSD calculations, ambiguity in correlations between P-Acc and Acc, questions about the necessity of the metric compared to existing measures like ROUGE, limited dataset testing, absence of human judgment, and missing details. During discussions, there was a strong concern regarding the lack of human judgment and the fact that the metric was not compared with baseline metrics. While the authors have shown new results during the response period, it was not possible to check for soundness.

---

### Decision · Program_Chairs · 2023-10-07

**Decision:**

Accept-Findings

**Comment:**

The paper introduces EGISES, a novel metric for evaluating personalization in summarization. Reviewers acknowledge the paper's theoretical contributions and its importance in addressing personalization in summarization, praising its clarity and organization. However, concerns include limited comparisons with classic metrics like BERTScore, computational complexity due to multiple JSD calculations, ambiguity in correlations between P-Acc and Acc, questions about the necessity of the metric compared to existing measures like ROUGE, limited dataset testing, absence of human judgment, and missing details. During discussions, there was a strong concern regarding the lack of human judgment and the fact that the metric was not compared with baseline metrics. While the authors have shown new results during the response period, it was not possible to check for soundness.